# Effectiveness of a Person-Centered Prescription Model in Hospitalized Older People at the End of Life According to Their Disease Trajectories and Frailty Index

**DOI:** 10.3390/ijerph20043542

**Published:** 2023-02-17

**Authors:** Alexander Ferro-Uriguen, Idoia Beobide-Telleria, Javier Gil-Goikouria, Petra Teresa Peña-Labour, Andrea Díaz-Vila, Arlovia Teresa Herasme-Grullón, Enrique Echevarría-Orella

**Affiliations:** 1Department of Pharmacy, Ricardo Bermingham Hospital—Matia Foundation, 20018 Donostia, Spain; 2Department of Physiology, University of the Basque Country (UPV/EHU), 48940 Bilbao, Spain; 3Network Centre for Biomedical Research in Mental Health to the Institute of Health Carlos III (CIBERSAM ISCIII), 28029 Madrid, Spain; 4Department of Geriatrics, Ricardo Bermingham Hospital—Matia Foundation, 20018 Donostia, Spain

**Keywords:** end of life, frail older adults, palliative medicine, deprescribing, patient-centered prescription model

## Abstract

This study aimed to comparatively analyze the effect of the person-centered prescription (PCP) model on pharmacotherapeutic indicators and the costs of pharmacological treatment between a dementia-like trajectory and an end-stage organ failure trajectory, and two states of frailty (cut-off point 0.5). A randomized controlled trial was conducted with patients aged ≥65 years admitted to a subacute hospital and identified by the Necessity of Palliative Care test to require palliative care. Data were collected from February 2018 to February 2020. Variables assessed included sociodemographic, clinical, degree-of-frailty, and several pharmacotherapeutic indicators and the 28-day medication cost. Fifty-five patients with dementia-like trajectory and 26 with organ failure trajectory were recruited observing significant differences at hospital admission in the mean number of medications (7.6 vs. 9.7; *p* < 0.004), the proportion of people on more than 10 medications (20.0% vs. 53.8%; *p* < 0.002), the number of drug–drug interactions (2.7 vs. 5.1; *p* < 0.006), and the Medication Regimen Complexity Index (MRCI) (25.7 vs. 33.4; *p* < 0.006), respectively. Also, regarding dementia-like patients, after application of the PCP model, these patients improved significantly in the intervention group compared to the control group in the mean number of chronic medications, STOPP Frail Criteria, MRCI and the 28-day cost of regular medications (*p* < 0.05) between admission and discharge. As for the PCP effect on the control and the intervention group at the end-stage organ failure, we did not observe statistically significant differences. On the other hand, when the effect of the PCP model on different degrees of frailty was evaluated, no unequal behavior was observed.

## 1. Introduction

The prevalence of advanced chronic diseases and the mortality of patients with a limited life expectancy are notably increasing. These are linked to dependency, frailty, and multimorbidity with different degrees of complexity, need, and demand [1,2]. Early identification of these individuals enables palliative care to be provided at the end of life (EOL) to help clarify treatment preferences and care goals to improve quality of life and symptom control, reduce distress, allow less aggressive care and a lower economic cost, and even prolong survival [3]. In many cases, this is not yet general practice, especially in patients with trajectories of non-oncological diseases [4,5], such as dementias, neurodegenerative diseases, or end-stage organ failure.

The quantification of a frailty index based on the model proposed by Rockwood et al. [6] defines frailty as a continuous variable that does not end with disability or dependency, as in the case of the identification of a frail phenotype [7]. Thus, this frailty index could help make a situational diagnosis and stratify patients to differentiate those with mild frailty, who can benefit from more preventive therapeutic interventions, from those with advanced frailty, for whom a more conservative approach is preferable [8].

However, in practice, the use of medications increases because these patients accumulate more deficits, leading to a higher degree of frailty and increasing the likelihood of a fatal outcome. This is due not only to the use of medications to alleviate the symptoms associated with the disease, but also to long-term preventive treatments of questionable clinical benefit [9,10]. In a nationwide cohort study of decedents who died from life-limiting conditions, drugs of questionable clinical benefit were commonly continued (32%) or even initiated (14%) during the last three months of life [11]. These proportions were highest among younger individuals (i.e., aged 75–84 years), people who died from organ failure, and those with many coexisting chronic conditions. Older people with advanced chronic diseases and limited life expectancy are frequently exposed to potentially inappropriate prescriptions and polypharmacy, leading to adverse health outcomes such as worsening quality of life, adverse drug reactions, falls, hospitalizations, and even death [11,12,13,14].

Polypharmacy and potentially inappropriate prescriptions increase healthcare spending, which affects both the patient and the health system, in addition to causing adverse health outcomes. A study on the frequency and cost of potentially inappropriate prescriptions for older people in Canada estimated that $75 per patient and annum was spent on potentially inappropriate medications outside of hospital settings [15]. Similarly, in a study conducted in Ireland, the total cost of potentially inappropriate prescribed drugs was 9% of the overall spending on drugs in older people (aged >70 years) [16]. Therefore, low-value practices compromise the efficiency and sustainability of the health system, in addition to risking patient safety.

The adequacy of pharmacotherapy is especially important in frail older people with limited life expectancy. This is often linked to the use of deprescription strategies in these patients [17]. However, although deprescribing is the planned, supervised dose reduction or stopping of a medication, this process is commonly obstructed by factors such as awareness (the prescriber’s insight into the appropriateness of their prescribing), inertia (failure to act despite awareness), self-efficacy (having the skills, knowledge, attitudes, and information to deprescribe), and feasibility (including resources, time, and medical culture influences) [18].

The person-centered prescription (PCP) model has been recently reported to reduce polypharmacy, potentially inappropriate prescriptions at EOL according to the Screening Tool of Older Persons’ Prescriptions in Frail adults with limited life expectancy (STOPPFrail) criteria [19], drug–drug interactions, anticholinergic and sedative load according to the Drug Burden Index (DBI) [20], therapeutic complexity according to the Medication Regimen Complexity Index (MRCI) [21], and the costs associated with pharmacological treatment in hospitalized patients in the final phase of life, maintaining the effect for three months after hospital discharge [22]. The main objective of this study was to conduct a comparative analysis of the effects of the PCP model on pharmacotherapeutic indicators and the costs associated with pharmacological treatment between the two main trajectories of non-cancerous disease—dementia-like trajectory and end-stage organ failure trajectory, and two states of frailty.

These two trajectories have been shown to be different in terms of many of the dimensions studied, such as functional, cognitive, nutritional, geriatric syndromes, and use of health resources; therefore, changes in the impact of the PCP model were expected.

## 2. Materials and Methods

### 2.1. Design

This study was a parallel-group, unblinded, randomized clinical trial conducted in a subacute hospital in Gipuzkoa, Spain. Participants were randomly selected to receive either the usual pharmaceutical care (control group) or an adapted PCP model (intervention group). The trial was registered with ClinicalTrials.gov (NCT05454644). Specifically, in this post hoc study, the effect of the PCP model on different disease trajectories has been compared, establishing two groups: (1) dementia-like trajectory, including patients with dementia or neurodegenerative diseases or those who, without being notable for an advanced chronic disease, had been identified as in the final phase of life; and (2) end-stage organ failure trajectory, including chronic pulmonary disease, chronic heart disease, serious chronic liver disease, and serious chronic renal disease. For the comparative analysis based on the state of frailty, the cut-off point was established as a Frail-VIG score of 0.5, arbitrarily based on previous studies [23].

All participants were aged ≥65 years and admitted to the geriatric convalescence unit, where they were identified according to their baseline in the first 24–72 h as having a non-oncological advanced chronic disease and needing palliative care, with a limited survival prognosis according to the Necessity of Palliative Care (NECPAL) test [24]. Patients with hospital stays of <72 h, as well as those transferred to other hospitals or units and imminently terminal patients, were excluded.

### 2.2. Randomization and Data Collection

Over 24 months (February 2018–February 2020), all patients with a positive NECPAL test who were admitted to the geriatric convalescence unit were selected consecutively and randomized to the study groups in a 1:1 ratio. Randomization was stratified by geriatrician. The independent variables included and collected from the computerized clinical records of the Basque Health Service (Osakidetza) and the computerized records of the subacute hospital were (1) sociodemographic characteristics comprising gender, age, marital status, type of coexistence, and Gijón socio-family assessment [25]; (2) clinical characteristics comprising advanced chronic disease category, Charlson Comorbidity Index [26], Frail-VIG [23], cognitive assessment according to the Global Deterioration Scale/Functional Assessment Staging (GDS/FAST) [27], functional assessment according to the Barthel Index, and the number of hospitalizations in the previous year; and (3) pharmacotherapeutic characteristics comprising the number of regular medications, number of patients with ≥10 regular medications (defined as excessive polypharmacy or hyperpolypharmacy), STOPPFrail criteria, DBI, total drug–drug interactions, MRCI, and the costs associated with pharmacological treatment in hospitalized patients in the final phase of life.

Drug treatment data and variables related to pharmacotherapy were collected from the primary care electronic prescriptions records of the Basque Health Service (Osakidetza) at hospital admission and discharge and during the study follow-up. Only regular prescriptions were recorded; those used on-demand or for a brief time were recorded separately. Lastly, the 28-day cost of prescriptions was estimated from the same source of drug treatment records and the December 2021 price list prescription (Nomenclator) of the Spanish Agency of Medicines and Health Products. Only the active prescriptions were considered at each time-point studied. Based on the retail price of each prescribed medication, the unit price in € corrected by the patient’s prescribed dose at each time-point was calculated. Subsequently, to calculate the 28-day cost of prescriptions, the corrected unit price was multiplied by 28.

### 2.3. Intervention

A PCP-EOL model was implemented [22] based on a previous model proposed by Espaulella et al. [28]. This model, conducted by a geriatrician and a clinical pharmacist, consisted of a systematic four-step process: (1) identify patients with an advanced chronic condition and limited life expectancy, (2) interview the patient or closest caregiver, (3) conduct a medication review, and (4) implement a treatment plan.

### 2.4. Usual Pharmaceutical Care

Essentially, this was based on medication reconciliation in the first 24–72 h of admission, as well as on validating medical prescription modifications during hospital stay. After randomization, the patients who received the usual pharmaceutical care were assigned to the control group.

### 2.5. Outcome Measures

The mean changes between admission and discharge were measured in the number of regular medications (as-needed medicines were not included, and combination products were included as one drug), STOPPFrail criteria, total drug–drug interactions, DBI, and MRCI. Any decrease in the pharmacotherapeutic variables studied during hospital admission was considered an optimization of the pharmacotherapy.

Likewise, the change in the 28-day cost of prescriptions in € was estimated between admission and discharge.

### 2.6. Statistical Analysis

The selected variables were expressed as mean, median, and frequency (percentages). Pearson’s χ^2^ test was used to compare the qualitative variables. Student’s *t*-test and the Mann–Whitney U-test were used to compare parametric and non-parametric distributions, respectively. Statistical analyses were performed using SPSS software (SPSS Inc., Chicago, IL, USA, version 20.0).

### 2.7. Ethical Considerations

The study (identity number: AFU-PPG-2017-01) was approved by the Clinical Research Ethics Committee of the Gipuzkoa Health Area. Informed consent was previously obtained from all recruited patients. In cases where participants had cognitive impairment, consent was obtained from legal guardians who acted as surrogate informants.

## 3. Results

Overall, 55 patients with dementia-like trajectory (T1; 22.5% of patients died) and 26 with end-stage organ failure trajectory (T2; 39.5% of patients died) were recruited. The study participants had a mean age of 87.3 ± 5.8 years, and 58% were female.

Table 1 details the differences between the baseline demographic, clinical, functional, and cognitive data according to illness trajectory. In patients with end-stage organ failure, a higher degree of comorbidity was observed, highlighting the diagnoses of congestive heart failure, chronic obstructive pulmonary disease, and moderate to severe chronic kidney disease. These patients manifested worse symptomatic control of dyspnea. These patients also had higher hospital attendance in the year before recruitment. In contrast, patients with a dementia-like trajectory reported a higher degree of functional and cognitive dependence.

As demonstrated in Table 2, patients with a more advanced frailty degree had more comorbidities, a greater proportion of comorbidities, and more advanced stages of dementia. Also notable was the greater presence of geriatric syndromes, such as depression, delirium, and malnutrition.

Differences were observed in the patterns of medication use between trajectories at hospital admission (Table 3). The mean number of medications (7.6 ± 2.8 vs. 9.7 ± 3.5), the proportion of people with more than 10 medications, the number of drug–drug interactions, and the complexity of the pharmacological treatments were higher in patients with end-stage organ failure. These differences were maintained between admission and discharge for certain pharmacotherapeutic variables, such as the number of medications, the proportion of people with more than 10 chronic medications, the STOPPFrail criteria, drug–drug interactions, and pharmacotherapeutic complexity. 

Likewise, the effect of the PCP model was greater in patients with a dementia-like trajectory. Studying the PCP effect for each of the trajectories in the control and intervention groups, we observed in dementia-like patients an improvement in the mean number of chronic medications, the STOPPFrail criteria, the pharmacotherapeutic complexity, and the 28-day cost of regular medications. Statistically significant differences were not observed in the PCP effect between the control and intervention groups in end-stage organ failure.

Analyzing the pharmacotherapeutic variables at admission according to frailty degree, no differences were observed (Table 4), except for DBI. Differences were found in anticholinergic and sedative load at both admission (0.9 ± 0.6 vs. 1.3 ± 0.8) and hospital discharge (0.8 ± 0.6 vs. 1.2 ± 0.7). On the other hand, no unequal behavior was observed when the effect of the PCP model on different degrees of frailty was evaluated.

## 4. Discussion

The two illness trajectories had a clearly different pattern of comorbidities at EOL, as well as in their functional and cognitive dependence. This is in accordance with the previous report by Lynn and Adamson [29]. Thus, people with end-stage organ failure present less functional and cognitive deterioration but stand out for greater hospital attendance before the identification of the EOL, in concordance with Amblas et al. [30]. Likewise, for this same illness trajectory, a high in-hospital mortality was observed (39.5%), indicating that the screening tools to offer palliative care at EOL could be applied too late for this patient profile.

Our results also demonstrate that patients with organ failure trajectories display greater polypharmacy, numbers of drug–drug interactions, and pharmacotherapeutic complexity. This means that this population is especially vulnerable to drug-related problems. These findings agree in part with those reported by Todd et al. [31], in which patients with a history of heart and lung disease presented a higher risk of exposure to inappropriate medications and a greater number of drug interactions. These interactions were often linked to a combination of drugs frequently used for the management of long-term clinical conditions.

To our knowledge, this is the first study to analyze the effect of a pharmacotherapeutic intervention on different illness trajectories at EOL, observing significant differences in the intensity of the effect in each of them. For patients with a dementia-like trajectory, the PCP model was effective in reducing the mean number of chronic medications, inappropriate or futile medications at EOL according to the STOPPFrail criteria, pharmacotherapeutic complexity, and the cost associated with medical prescriptions. However, the effect on organ failure trajectories was more modest and not significant. This lower effect in patients with end-stage organ failure may be related to difficulties involved in formulating a short- to medium-term prognosis and identifying the terminal phase of such diseases, concluding in the so-called prognostic paralysis [4].

The analysis performed according to the degree of frailty of patients at EOL reported a high proportion of patients with an advanced stage (Frail-VIG > 0.5; 53%), characterized by high comorbidity, and up to 83.7% of people had a diagnosis of any degree of dementia. Notably, in previous studies, patients with advanced frailty presented a mortality rate at one–two years of practically 100% [23], so it could be expected that the PCP model could be more effective in those with a more advanced stage. However, in our case, for both patients with mild–moderate frailty and those with advanced frailty, the PCP model was effective for most pharmacotherapeutic indicators. The degree of frailty was not a differential element in the intensity of the model’s effect. Nevertheless, notably, a recently published before–after study reported that the application of a PCP model was valid for reducing polypharmacy, MRCI, and DBI in patients with a moderate or advanced degree of frailty and multimorbidity but who were not necessarily at EOL [32]. However, in this last case, the results could be because people with a more advanced frailty index were entering a final stage of life, whereas the PCP model has greater efficacy when optimizing pharmacotherapy.

In addition, the comparative analysis of the baseline pharmacotherapeutic characteristics between different degrees of frailty highlighted a higher DBI in those with advanced frailty. This may be related to a situation closer to the EOL, which requires these patients to use more drugs with anticholinergic and sedative characteristics [33,34,35]. Notably, greater exposure to drugs with anticholinergic activity is associated with more fatigue, dry mouth, worse concentration, and worsening status at EOL [34]. Further research on this is needed to clarify whether anticholinergic load directly causes this worsening status or whether people who are worsening need more medication to optimize their state of comfort.

In people with a dementia-like trajectory, the PCP model has improved pharmacotherapeutic indicators, leading to a decrease in the monthly cost of medications. From the perspective of health economics, pharmaceutical spending may represent only a small part of the total healthcare spending—15% in the case of Spain [36]. Nevertheless, the improvement of pharmacotherapeutic indicators could reduce the costs caused by new hospitalizations or referrals to long-term care homes.

However, the effect on the cost of pharmacotherapy for people with an organ failure trajectory was not significant. Importantly, people with a history of primarily advanced heart or lung disease are those who are associated with a higher health care cost in the last 12 months of life, due to 80% being readmitted to hospital [37]. Thus, the PCP model for this trajectory must be optimized to improve pharmacotherapeutic indicators and analyzed for the impact it may have on the reduction of direct and indirect costs (e.g., hospital readmissions) by pharmacotherapy, which will ultimately result in an improvement in quality of life.

This study has some limitations. First, the objectives were based on surrogate variables such as pharmacotherapeutic indicators and not on health outcomes that could facilitate the interpretation of the results’ clinical significance. Second, the economic analysis considered only the savings on direct costs associated with pharmacological prescriptions. Future studies should also consider savings on indirect costs, such as decreased medical consultations or hospital readmissions caused by therapy optimization.

However, this work also has notable strengths. Establishing the PCP model in a hospital’s geriatric convalescence unit has allowed systematizing the early identification of EOL in people with trajectories of non-oncological diseases, guaranteeing universal coverage of palliative care. It has also allowed an understanding of the effectiveness of the PCP model on the different disease trajectories and frailty statuses.

## 5. Conclusions

The reported PCP model has proven more effective, improving pharmacotherapeutic indicators and the cost of prescriptions for people with a dementia-like trajectory in comparison to those with an end-stage organ failure trajectory. The former group has the potential to protocolize this model of pharmacotherapeutic adequacy. In contrast, for people with an end-stage organ failure trajectory, the effectiveness was moderate, reducing the presence of STOPPFrail criteria, although in no case did the pharmacotherapeutic parameters worsen. This indicates the necessity to understand the main barriers for physicians, patients, and family members that prevent these from being as effective as for other disease trajectories, as well as the need to assess a modification of the protocol for the latter trajectory. Optimizing the early identification of palliative care will also be necessary, as well as greater scientific evidence on deprescription strategies for this trajectory that allow for more effective intervention and less uncertainty.

Once the need for palliative care was identified using the NECPAL screening tool, the frailty degree was not a critical element that discriminated against the effectiveness of the PCP model. This allows it to be an effective model in people with both mild–moderate and advanced frailty.

## Figures and Tables

**Table 1 ijerph-20-03542-t001:** Cohort baseline data according to illness trajectory.

Variable	T1 (*n* = 55)	T2 (*n* = 26)	*p*
Women, *n* (%)	31 (56.4)	16 (61.5)	0.660
Mean age, years (SD)	87.7 (5.6)	86.6 (6.3)	0.443
Marital status, *n* (%)			0.971
- Unmarried, divorced, separated	7 (12.7)	3 (11.5)	
- Married	22 (40.0)	10 (38.5)	
- Widowed	26 (47.3)	12 (50.0)	
Type of coexistence, *n* (%)			0.619
- Alone	10 (18.2)	3 (11.5)	
- Spouse	22 (40.0)	10 (38.5)	
- Children or other relatives	14 (25.5)	10 (38.5)	
- Other caregivers	9 (16.4)	3 (11.5)	
Gijón’s socio-family assessment, media (SD)	12.3 (2.6)	11.8 (2.5)	0.433
Gijón’s socio-family assessment, *n* (%)			0.436
- Good Social Status (0–9 points)	8 (14.6)	4 (15.4)	
- Social Risk (10–14 points)	34 (61.8)	19 (73.1)	
- Social Problem (≥15 points)	13 (23.6)	3 (11.5)	
Place of provenance, *n* (%)			0.730
- Hospital	50 (90.9)	23 (88.5)	
- Primary care/nursing home	5 (9.1)	3 (11.5)	
CCI, median (IQR)	3 (4)	4 (2)	0.001 *
No. of patients with ≥ 3 points CCI, *n* (%)	33 (60.0)	23 (88.5)	0.010 *
Diagnoses, *n* (%)			
- Myocardial infarction	7 (12.7)	7 (26.9)	0.115
- Congestive heart failure	15 (27.3)	20 (76.9)	<0.001 *
- Peripheral vascular disease	4 (7.3)	4 (15.4)	0.253
- Cerebrovascular accident	17 (30.9)	8 (30.8)	0.990
- Dementia	37 (67.3)	11 (42.3)	0.033 *
- Chronic obstructive pulmonary disease	4 (7.3)	12 (46.2)	<0.001 *
- Diabetes mellitus	19 (34.5)	12 (46.2)	0.316
- Moderate to severe chronic kidney disease	18 (32.7)	16 (61.5)	0.014 *
- Cancer without metastases	10 (18.2)	4 (15.4)	0.756
GDS ≥ 6, *n* (%)	26 (47.3)	6 (23.1)	0.038 *
Barthel Index, median (IQR)	32 (47)	67 (60)	0.003 *
Barthel Index ≤ 35, *n* (%)	34 (61.8)	10 (38.5)	0.049 *
Frail-VIG, mean (SD)	0.51 (0.11)	0.51 (0.12)	0.887
Frail-VIG > 0.50, *n* (%)	28 (50.9)	15 (57.7)	0.568
Geriatric Syndromes, *n* (%)			
- Depressive syndrome	27 (49.1)	7 (26.9)	0.059
- Insomnia/anxiety ^a^	39 (70.9)	18 (69.2)	0.877
- Delirium ^b^	18 (32.7)	6 (23.1)	0.375
- Falls ^c^	25 (45.5)	6 (23.1)	0.053
- Pressure ulcers	13 (23.6)	4 (15.4)	0.395
- Dysphagia	27 (49.1)	5 (19.2)	0.010 *
- Malnutrition (≥5% weight loss in the last 6 months)	20 (36.4)	6 (23.1)	0.232
Mayor symptoms			
- Pain	8 (14.5)	4 (15.4)	0.921
- Dyspnoea	2 (3.6)	8 (30.8)	0.001 *
No. of hospitalizations, median (IQR)	1 (1)	1 (3)	0.040 *
Hospitalizations in the last year, *n* (%)			0.001 *
- 0	25 (45.5)	7 (26.9)	
- 1	18 (32.7)	8 (30.8)	
- ≥2	12 (21.8)	11 (42.3)	

T1, Dementia-like trajectory; T2, End-stage organ failure trajectory; CCI, Charlson Comorbidity Index; DBI, Drug Burden Index; Frail-VIG, Frailty Index based on Comprehensive Geriatric Assessment; GDS, Reisberg’s Global Deterioration Scale; IQR, Interquartile Range; MRCI, Medication Regimen Complexity Index; PIM, potentially inappropriate medications; SD, standard deviation. ^a^ Need benzodiazepines or other psychotropics profile sedative for insomnia/anxiety. ^b^ Delirium or behavioral disorder that has required taking neuroleptics in the last six months. ^c^ ≥2 falls or a fall requiring hospitalization in the last six months. * *p* < 0.05.

**Table 2 ijerph-20-03542-t002:** Cohort baseline data according to frailty degree.

Variable	Frail-VIG ≤ 0.5 (*n* = 38)	Frail-VIG > 0.5 (*n* = 43)	*p*
Women, *n* (%)	22 (57.9)	25 (58.1)	0.982
Mean age, years (SD)	88.5 (5.8)	86.3 (5.7)	0.088
Marital status, *n* (%)			0.662
- Unmarried, divorced, separated	6 (15.8)	4 (9.3)	
- Married	14 (36.8)	18 (41.9)	
- Widowed	18 (47.4)	21 (48.8)	
Type of coexistence, *n* (%)			0.527
- Alone	8 (21.1)	5 (11.6)	
- Spouse	14 (36.8)	18 (41.9)	
- Children or other relatives	12 (31.6)	12 (27.9)	
- Other caregivers	4 (10.5)	8 (18.6)	
Gijón’s socio-family assessment, media (SD)	12.1 (2.5)	12.2 (2.7)	0.825
Gijón’s socio-family assessment, *n* (%)			0.921
- Good Social Status (0–9 points)	6 (15.7)	6 (14.0)	
- Social Risk (10–14 points)	24 (63.2)	29 (67.4)	
- Social Problem (≥15 points)	8 (21.1)	8 (18.6)	
Place of provenance, *n* (%)			0.191
- Hospital	36 (94.7)	37 (86.0)	
- Primary care/nursing home	2 (5.3)	6 (14.0)	
CCI, median (IQR)	3 (4)	4 (2)	0.046 *
No. of patients with ≥ 3 points CCI, *n* (%)	22 (57.9)	34 (79.1)	0.040 *
Diagnoses, *n* (%)			
- Myocardial infarction	4 (10.5)	10 (23.3)	0.131
- Congestive heart failure	14 (36.8)	21 (48.8)	0.277
- Peripheral vascular disease	3 (7.9)	5 (11.6)	0.574
- Cerebrovascular accident	11 (28.9)	14 (32.6)	0.726
- Dementia	12 (31.6)	36 (83.7)	<0.001 *
- Chronic obstructive pulmonary disease	6 (15.8)	10 (23.3)	0.400
- Diabetes mellitus	16 (42.1)	15 (34.9)	0.505
- Moderate to severe chronic kidney disease	19 (50.0)	15 (34.9)	0.169
- Cancer without metastases	5 (13.2)	9 (20.9)	0.356
GDS ≥ 6, *n* (%)	8 (21.1)	24 (55.8)	0.001 *
Barthel Index, median (IQR)	49 (51)	34 (66)	0.151
Barthel Index ≤ 35, *n* (%)	17 (44.7)	27 (62.8)	0.104
Geriatric Syndromes, *n* (%)			
- Depressive syndrome	10 (26.3)	24 (55.8)	0.007 *
- Insomnia/anxiety ^a^	18 (47.4)	39 (90.7)	<0.001
- Delirium ^b^	3 (7.9)	21 (48.8)	<0.001 *
- Falls ^c^	12 (31.6)	19 (44.2)	0.244
- Pressure ulcers	6 (15.8)	11 (25.6)	0.280
- Dysphagia	12 (31.6)	20 (46.5)	0.170
- Malnutrition (≥5% weight loss in the last 6 months)	8 (21.1)	18 (41.9)	0.045 *
Mayor symptoms			
- Pain	6 (15.8)	6 (14.0)	0.816
- Dyspnoea	5 (13.2)	5 (11.6)	0.835
No. of hospitalization, median (IQR)	1 (2)	1 (2)	0.909
Hospitalization in the last year, *n* (%)			0.993
- 0	15 (39.5)	17 (39.5)	
- 1	12 (31.6)	14 (32.6)	
- ≥2	11 (28.9)	12 (27.9)	

Frail-VIG, Frailty Index based on Comprehensive Geriatric Assessment; CCI, Charlson Comorbidity Index; DBI, Drug Burden Index; GDS, Reisberg’s Global Deterioration Scale; IQR, Interquartile Range; MRCI, Medication Regimen Complexity Index; PIM, potentially inappropriate medications; SD, standard deviation. ^a^ Need benzodiazepines or other psychotropics profile sedative for insomnia/anxiety. ^b^ Delirium or behavioral disorder that has required taking neuroleptics in the last six months. ^c^ ≥2 falls or a fall requiring hospitalization in the last six months. * *p* < 0.05.

**Table 3 ijerph-20-03542-t003:** Admission and discharge review analysis according to illness trajectory.

Variable		T 1	T 2	*p*	T 1		*p*	T 2		*p*
		All (*n* = 55)	All (*n* = 26)		CG (*n* = 27)	IG (*n* = 28)		CG (*n* = 15)	IG (*n* = 11)	
**No. of regular medications, mean (SD)**	Admission	7.6 (2.8)	9.7 (3.5)	0.004 *	7.4 (2.5)	7.7 (3.1)	0.760	9.1 (3.9)	10.4 (2.8)	0.353
	Discharge	6.4 (2.4)	9.4 (3.6)	<0.001 *	7.1 (2.5)	5.7 (2.2)	0.032 *	9.5 (4.4)	9.3 (2.4)	0.861
	Difference	−1.2 (2.5)	−0.3 (3.0)	0.162	−0.3 (2.2)	−2.0 (2.5)	0.013 *	0.4 (2.7)	−1.1 (3.3)	0.196
**No. of patients with ≥ 10 regular medications, *n* (%)**	Admission	11 (20.0)	14 (53.8)	0.002 *	5 (18.5)	6 (21.4)	0.787	6 (40.0)	8 (72.7)	0.098
	Discharge	5 (9.1)	11 (42.3)	<0.001 *	4 (14.8)	1 (3.6)	0.147	7 (46.7)	4 (36.4)	0.599
**STOPP Frail-defined PIMs at the EOL, mean (SD)**	Admission	1.6 (1.2)	1.9 (1.6)	0.350	1.4 (1.1)	1.8 (1.3)	0.306	2.3 (1.7)	1.5 (1.5)	0.214
	Discharge	0.5 (0.8)	1.2 (1.5)	0.047 *	0.9 (0.9)	0.1 (0.4)	<0.001 *	1.9 (1.7)	0.2 (0.4)	0.001 *
	Difference	−1.1 (1.2)	−0.7 (1.1)	0.207	−0.5 (0.8)	−1.7 (1.3)	<0.001 *	−0.4 (0.7)	−1.3 (1.3)	0.054
**DBI, mean (SD)**	Admission	1.17 (0.79)	1.04 (0.71)	0.472	1.10 (0.73)	1.24 (0.85)	0.517	0.83 (0.61)	1.33 (0.78)	0.075
	Discharge	0.97 (0.36)	1.11 (0.67)	0.362	1.01 (0.68)	0.92 (0.66)	0.629	1.04 (0.57)	1.22 (0.81)	0.509
	Difference	−0.20 (0.50)	0.07 (0.61)	0.032 *	−0.09 (0.33)	−0.32 (0.60)	0.088	0.21 (0.51)	−0.11 (0.72)	0.191
**Total Drug–Drug Interactions, mean (SD)**	Admission	2.7 (2.5)	5.1 (3.8)	0.006 *	2.6 (2.3)	2.9 (2.7)	0.585	4.5 (3.9)	6.0 (3.5)	0.315
	Discharge	1.9 (1.8)	4.7 (4.5)	0.005 *	2.2 (1.8)	1.6 (1.8)	0.248	5.3 (5.0)	3.9 (3.7)	0.456
	Difference	−0.8 (2.0)	−0.4 (4.6)	0.679	−0.4 (2.3)	−1.3 (1.7)	0.082	0.8 (5.2)	−2.1 (3.1)	0.116
**MRCI, mean (SD)**	Admission	25.7 (10.4)	33.4 (13.6)	0.006 *	24.3 (9.3)	27.0 (11.4)	0.348	32.3 (16.4)	35.0 (9.2)	0.618
	Discharge	22.8 (9.9)	33.6 (12.8)	<0.001 *	25.6 (10.7)	20.1 (8.4)	0.039 *	34.3 (15.6)	32.7 (8.0)	0.763
	Difference	−2.9 (9.9)	0.19 (9.1)	0.185	1.3 (9.7)	−6.9 (8.5)	0.002 *	2.0 (7.9)	−2.3 (10.4)	0.239
**28-day cost of regular medications, mean (SD)**	Admission	101.5 (51.3)	113.5 (83.0)	0.426	103.0 (56.1)	100.1 (62.2)	0.834	88.6 (54.5)	123.3 (70.5)	0.179
	Discharge	81.0 (56.4)	103.9 (60.9)	0.100	101.0 (62.2)	61.7 (42.8)	0.008 *	91.2 (55.3)	120.2 (69.8)	0.259
	Difference	20.5 (48.4)	9.6 (62.6)	0.392	2.0 (46.2)	−38.4 (44.2)	0.004 *	2.6 (25.0)	−3.16 (53.8)	0.760

T1, Dementia-like trajectory; T2, End-stage organ failure trajectory; CG, Control Group; IG, Intervention Group; DBI, Drug Burden Index; EOL, End Of Life; MRCI, Medication Regimen Complexity Index; PIM, potentially inappropriate medications; SD, standard deviation. * *p* < 0.05.

**Table 4 ijerph-20-03542-t004:** Admission and discharge review analysis according to frailty degree.

Variable		F-VIG ≤ 0.5	F-VIG > 0.5		F-VIG ≤ 0.5			F-VIG > 0.5		
		All (*n* = 38)	All (*n* = 43)	*p*	CG (*n* = 22)	IG (*n* = 16)	*p*	CG (*n* = 20)	IG (*n* = 23)	*p*
**No. of regular medications, mean (SD)**	Admission	8.0 (3.6)	8.4 (2.8)	0.609	7.5 (3.2)	8.7 (4.0)	0.311	8.6 (3.0)	8.3 (2.7)	0.701
	Discharge	7.4 (3.2)	7.3 (3.2)	0.894	7.6 (3.3)	7.1 (3.1)	0.634	8.3 (3.6)	6.4 (2.5)	0.049 *
	Difference	−0.6 (2.6)	−1.1 (2.7)	0.443	0.1 (2.0)	−1.6 (3.1)	0.047 *	−0.2 (2.7)	−1.8 (2.5)	0.057
**No. of patients with ≥ 10 regular medications, *n* (%)**	Admission	11 (28.9)	14 (32.6)	0.726	5 (22.7)	6 (37.5)	0.321	6 (30.0)	8 (24.8)	0.739
	Discharge	7 (18.4)	9 (20.9)	0.777	4 (18.2)	3 (18.8)	0.964	7 (35.0)	2 (8.7)	0.034 *
**STOPP Frail-defined PIMs at the EOL, mean (SD)**	Admission	1,8 (1.5)	1.7 (1.2)	0.772	1.8 (1.4)	1.7 (1.6)	0.795	1.6 (1.3)	1.7 (1.2)	0.906
	Discharge	0.8 (1.1)	0.7 (1.1)	0.868	1.2 (1.3)	0.1 (0.3)	0.001 *	1.3 (1.4)	0.2 (0.4)	<0.001 *
	Difference	−1.0 (1.2)	−0.9 (1.2)	0.862	−0.6 (0.7)	−1.6 (1.5)	0.027 *	−0.3 (0.86)	−1.5 (1.2)	<0.001*
**DBI, mean (SD)**	Admission	0.9 (0.6)	1.3 (0.8)	0.005 *	0.81 (0.55)	0.97 (0.70)	0.418	1.22 (0.79)	1.47 (0.86)	0.321
	Discharge	0.8 (0.6)	1.2 (0.7)	0.005 *	0.78 (0.62)	0.82 (0.53)	0.831	1.29 (0.55)	1.14 (0.80)	0.483
	Difference	−0.1(0.3)	−0.1 (0.7)	0.610	−0.03 (0.23)	−0.15 (0.42)	0.294	0.07 (0.57)	−0.33 (0.75)	0.056
**Total Drug–Drug Interactions, mean (SD)**	Admission	3.3 (3.5)	3.7 (2.7)	0.612	2.6 (3.5)	4.3 (3.5)	0.145	3.9 (2.4)	3.4 (3.0)	0.546
	Discharge	2.8 (3.0)	2.8 (3.4)	0.947	2.8 (2.6)	2.8 (3.6)	0.970	3.9 (4.4)	1.9 (1.7)	0.072
	Difference	−0.5 (2.5)	−0.8 (3.5)	0.653	0.2 (2.3)	−1.5 (2.6)	0.043 *	−0.0 (4.7)	−1.5 (1.8)	0.172
**MRCI, mean (SD)**	Admission	26.7 (12.9)	29.5 (11.2)	0.305	25.0 (12.6)	29.1 (13.3)	0.349	29.5 (12.6)	29.4 (10.0)	0.974
	Discharge	26.3 (11.0)	26.3 (12.9)	0.978	27.1 (11.7)	25.3 (10.2)	0.623	30.5 (14.7)	22.5 (10.0)	0.049 *
	Difference	−0.4 (8.6)	−3.2 (10.6)	0.192	2.1 (7.9)	−3.8 (8.5)	0.036 *	0.9 (10.3)	−6.9 (9.5)	0.014 *
**28-day cost of regular medications, mean (SD)**	Admission	111.8 (75.9)	99.6 (49.0)	0.400	98.5 (60.5)	130.2 (92.0)	0.208	97.2 (50.4)	101.8 (48.8)	0.764
	Discharge	90.3 (54.4)	86.7 (62.4)	0.779	92.8 (61.2)	86.9 (45.2)	0.746	102.7 (58.5)	72.7 (63.7)	0.118
	Difference	−21.5 (59.0)	−13.0 (47.8)	0.476	−5.7 (41.6)	−43.3 (72.8)	0.051	5.5 (37.5)	−29.1 (50.6)	0.016 *

Frail-VIG, Frailty Index based on Comprehensive Geriatric Assessment; CG, Control Group; IG, Intervention Group; DBI, Drug Burden Index; EOL, End Of Life; MRCI, Medication Regimen Complexity Index; PIM, potentially inappropriate medications; SD, standard deviation. * *p* < 0.05.

## Data Availability

The datasets generated during and/or analyzed during the current study are available from the corresponding author on reasonable request.

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
