# Peer review of "Effectiveness of a Person-Centered Prescription Model in Hospitalized Older People at the End of Life According to Their Disease Trajectories and Frailty Index"

_ijerph, 2023, doi:10.3390/ijerph20043542_

Round 1

Reviewer 1 Report

It is an excellent work and well executed and well written. Minor spell check required. 

Author Response

Many thanks for your comments. Taking your feedback into account, the spelling has been corrected in the new revised manuscript.

Reviewer 2 Report

Overall, the manuscript is well presented. However, several parts can be improved. Please provide more detail information regarding gap of study, sampling, intervention and data analysis in the abstract. In addition, please add explanation regarding research problem in the introduction, explain how the researchers determine the sample size, more detail information about intervention, the validity and reliability of the tools.

Author Response

Authors response:

We are very grateful to Reviewer 2 for your help in improving our manuscript. Taking into account your comments and recommendations, point by point response is give below:

Abstract has been modified in the new revised manuscript providing more detail about sampling, intervention and data analysis.

“Abstract: This study aimed to comparatively analyze the effect of the person-centered prescription (PCP) model on pharmacotherapeutic indicators and the costs of pharmacological treatment between a dementia-like trajectory and an end-stage organ failure trajectory, and two states of frailty (cut-off point 0.5). A randomized controlled trial was conducted with patients aged ≥65 years admitted to a subacute hospital and identified by the Necessity of Palliative Care test to require palliative care. Data were collected from February 2018 to February 2020. Variables assessed included sociodemographic, clinical, degree of frailty , pharmacotherapeutic indicators  and 28-day medication cost). 55 patients with dementia-like trajectory and 26 with organ failure trajectory were recruited observing significant differences at hospital admission in the mean number of medications (7.6 vs. 9.7; p<0.004), the proportion of people on more than 10 medications (20.0% vs. 53.8%; p<0.002), the number of drug-drug interactions (2.7 vs. 5.1; p<0.006), and the Medication Regimen Complexity Index (25.7 vs. 33.4; p<0.006), respectively. Also, regarding dementia-like patients, after application of PCP model improved significantly in the intervention group compared to the control group in the mean number of chronic medications, STOPP Frail Criteria, MRCI and the 28-day cost of regular medications (p<0.05) between admission and discharge. As for the PCP effect on the control and the intervention group at the end-stage organ failure, we did not observe statistically significant differences. On the other hand, when the effect of the PCP model on different degrees of frailty was evaluated, no unequal behavior was observed.”

Moreover, the introduction section has been modified as follows in the new revised manuscript:

Page 2, line 99: “These two trajectories have been shown to be different in terms of many of the dimensions studied, such as functional, cognitive, nutritional, geriatric syndromes, and use of health resources, therefore changes in the impact of the PCP model were expected.”

The sample size. This is a post hoc study of a previous research, whose objective was to assess the effectiveness of a person-centered prescription model in hospitalized older people at the end of life. In the present study, the impact of the PCP model between different disease trajectories and between different degrees of frailty has been studied. Therefore, given the post-hoc nature of the study, a predetermination of the sample size was not been carried out. However, a brief description has been added in the study design section in the new revised manuscript:

Page 3, line 122: “Specifically, in this post hoc study, the effect of the PCP model on different disease trajectories has been compared, establishing two groups”

Information about intervention, as well as the validity and reliability. We agree with the referee in the importance of detailing  model application. However, the PCP-EOL model was recently published in a full detail by our research team and has been cited in section 2.3. of the article (reference 22). This article briefly explains the 4 main steps to follow to apply the new PCP model and avoid redundancy on an already published topic.  

Reviewer 3 Report

Thank you very much for the opportunity to review the manuscript titled: Effectiveness of a person-centered prescription model in hospitalized older people at the end of life according to their disease trajectories and frailty index

The work is generally interesting, however below are my comments:

1.  Please define polypharmacy.  The most commonly reported definition of polypharmacy was the numerical definition of five or more medications daily.  Explain in the methodology why you choose 10 regular medications

2. Please describe the model PCP

3. The main objective of your study was to conduct a comparative analysis of the effects of the PCP model on pharmacotherapeutic indicators and the costs associated with pharmacological treatment between the two main trajectories of non-cancerous disease – dementia-like trajectory and end-stage organ failure trajectory, and two states of frailty.

However, the methodology section results did not show the results of this intervention. Rather there are results between illness trajectory and frailty

4. Please describe the usual pharmaceutical care

5. Please check Table 1a Type of coexistence and Gijón’s socio-family assessment - % are not exactly 100; Table 1b - Gijón’s socio-family assessment - % is not exactly 100

6. Conclusions
The reported PCP model has proven more effective, improving pharmacotherapeutic 307 indicators and the cost of prescriptions for people with a dementia-like trajectory in com-308 parison to those with end-stage organ failure trajectory - in the results there is no comparison to the group the usual pharmaceutical care

Author Response

We are very grateful to Reviewer 3 for your help in improving our manuscript. Taking into account your comments and recommendations, point by point response is give below:

  1. Please define polypharmacy. The most commonly reported definition of polypharmacy was the numerical definition of five or more medications daily.  Explain in the methodology why you choose 10 regular medications

Authors response: We agree with referee in that the term polypharmacy is usually refered to the use of five or more medications. Howewer, in this case, the term hyperpolypharmacy was used due to the fact that this population displayed very high average number of medications (90% consumed more than five medications).

Moreover, following the referee´s suggestion, the sentence defined as excessive polypharmacy or hyperpolypharmacy” has been included in the new revised manuscript (page 4, line 152).

  1. Please describe the model PCP

Authors response: We agree with the referee in the importance of detailing  model application. However, the PCP-EOL model was recently published in a full detail by our research team and has been cited in section 2.3. of the article (reference 22). This article briefly explains the 4 main steps to follow to apply the new PCP model and avoid redundancy on an already published topic. 

  1. Please describe the usual pharmaceutical care:

Authors response: Following the referee´s suggestion, the sentence “Essentially, it was based on medication reconciliation in the first 24-72h of admission, as well as on validating medical prescriptions modifications during hospital stay. After randomization, the patients who received the usual pharmaceutical care were assigned to the control group.” has been included.

  1. Please check Table 1a Type of coexistence and Gijón’s socio-family assessment - % are not exactly 100; Table 1b - Gijón’s socio-family assessment - % is not exactly 100

Authors response: The table 1 has been corrected in the new revised manuscript.

  1. The main objective and Conclusions.

The main objective of your study was to conduct a comparative analysis of the effects of the PCP model on pharmacotherapeutic indicators and the costs associated with pharmacological treatment between the two main trajectories of non-cancerous disease – dementia-like trajectory and end-stage organ failure trajectory, and two states of frailty.

However, the methodology section results did not show the results of this intervention. Rather there are results between illness trajectory and frailty

Conclusions.The re ported PCP model has proven more effective, improving  pharmacotherapeutic 307 indicators and the cost of prescriptions for people with a dementia-like trajectory in com-308 parison to those with end-stage organ failure trajectory - in the results there is no comparison to the group the usual pharmaceutical care.

Authors response: The previous submitted manuscript did not specify that the people who received the usual pharmaceutical care belonged to the control group. Once correctly detailed in the methodology (page 3, line 121), it should be understood as follows:

First, it is compared whether there are differences in the pharmacotherapeutic and cost indicators of patients belonging to one trajectory or another after a stay in a subacute hospital. However, in a subsequent analysis, each trajectory is analyzed separately to compare in this case the effect of PCP model (Intervention Group) and usual pharmaceutical care (Control Group).